# Regulation of Gut Microbiota through Breast Milk Feeding Benefits Language and Cognitive Development of Preterm Toddlers

**DOI:** 10.3390/microorganisms11040866

**Published:** 2023-03-28

**Authors:** Shan Guo, Kaikun Huang, Ruixia Liu, Jing Sun, Chenghong Yin

**Affiliations:** 1Department of Central Laboratory, Beijing Obstetrics and Gynecology Hospital, Capital Medical University, Beijing 100026, China; 2Institute for Integrated and Intelligent Systems, School of Medicine and Dentistry, Griffith University, Gold Coast, QLD 4215, Australia

**Keywords:** breast milk, preterm toddler, gut microbiota, neurodevelopment outcome

## Abstract

Feeding practice is essential to growth and development of preterm toddlers. However, the relationship of feeding mode with gut microbiota and neurodevelopment outcomes of preterm toddlers has not been characterized fully. We conducted this cohort study to assess neurodevelopment outcomes and gut microbiota community structures of preterm toddlers who received either breast milk, formula or mixed feeding. Fifty-five preterm toddlers born <37 weeks and 24 term toddlers were recruited in the study. Bayley III mental and physical index scores were measured among preterm toddlers at 12 ± 2 and 18 ± 2 months corrected age (CA). Gut microbiome composition was analyzed by 16S rRNA gene sequencing in fecal samples collected from all participants at 12 months, 16 months and 20 months after birth. We found exclusive breast milk feeding for over three months in the first six months after birth was associated with significant increase in language composite score at 12 months CA (86 (79,97) vs. 77 (71.75,79), *p* = 0.008) and both language (106.05 ± 14.68 vs. 90.58 ± 12.25, *p* = 0.000) and cognitive composite score at 18 months CA (107.17 ± 10.85 vs. 99.00 ± 9.24, *p* = 0.007). The alpha diversity, beta diversity and composition of gut microbiota from those breastfed preterm toddlers not only resembled healthy term toddlers but also followed similar structure of preterm toddlers with enhanced language and cognitive performance. Our results suggest exclusive breast milk feeding for over three months in preterm toddlers leads to optimal cognitive and language development and well-balanced microbiota.

## 1. Introduction

Preterm infant is defined as gestation age less than 37 weeks. An increasing number of preterm infants are surviving due to advanced treatment techniques in neonatal intensive care units (NICU) in recent decades. Nevertheless, rates of adverse neurodevelopmental outcome increased with decrease in gestational age that brings early life nutrition and subsequence neurodevelopment to the forefront [1].

Feeding options play vital roles in early growth and neural development of toddlers. The relationship of early and exclusive breast milk feeding with reduction of neonatal co-morbidities is evident and breast milk is considered the superior food for richer nutrients such as long-chain polyunsaturated fatty acids (LC-PUFA), contributing to neurogenesis for those who were born full term [2]. However, if unsupplemented breast milk cannot meet the nutritional demands of preterm toddlers [3], then nutrient-enriched formula enhances early growth of preterm infants, which possibly improves neurodevelopmental outcome through preventing growth failure and postnatal malnutrition from happening [4]. The finding of a large population prospective cohort in 11 countries in Europe demonstrated that breast milk feeding was associated with better cognitive development of very preterm toddlers at 2 years CA [5], but other studies showed conflicting conclusions [6,7].

Emerging evidence suggests that gut microbiota affects brain function through the bidirectional communication of the microbiome-gut-brain axis. Studies of animals raised in several conditions such as germ-free and usage of antibiotics associated with the alternation of early postnatal gut microbiome identified abnormalities in brain structure and brain function. These effects can endure into later childhood and adulthood, which emphasizes the significance of healthy microbiome during key neurodevelopmental phases [8]. Although early life dietary events have a significant role in development of infant gut microbiota, few studies associated diet-related microbiota pattern with neurodevelopment outcomes simultaneously, especially in preterm infants, this fragile group being more vulnerable to neurodevelopment defect.

In response to the lack of published results and also valid understanding of the impact of feeding practice on preterm infants, the aim of this cohort study is to figure out: (1) whether neurodevelopmental outcomes at 12 and 18 months CA of preterm toddlers receiving exclusive breast milk feeding initiated after discharge for over three months during first 6 months after birth differ from those who received formula or mixed feeding; and (2) whether diet-related difference in neurodevelopmental outcomes is connected with diet-related microbiota pattern. It is hypothesized that breast milk feeding would be associated with enhanced neurodevelopmental scores and healthier microbiota structure in preterm toddlers.

## 2. Materials and Methods

### 2.1. Study Subjects

This cohort study was conducted in Beijing Obstetrics and Gynecology Hospital, Capital Medical University, from September 2019 to December 2021. Fifty-five preterm toddlers (gestational age < 37 weeks) and 24 term toddlers (gestational age ≥ 37 weeks) born between September 2019 and May 2020 were included in the study. All participants were recruited from The China birth cohort (CBC). The study protocol was approved by The Ethics Committee of the Beijing Obstetrics and Gynecology Hospital. All parents of toddlers signed informed consent forms. Toddlers were excluded if: (1) family members had history of mental diseases; (2) toddlers had hereditary diseases or birth defects that affected neurodevelopment; and (3) toddlers had operation histories during follow up.

### 2.2. Data Collection

Pregnancy related information and neonatal clinical data were extracted from medical charts as follows: gestational age, birth weight, length, head circumference, gender, 1 min Apgar score, 5 min Apgar score, 10 min Apgar score, hospitalization days, delivery mode, antibiotic use duration, neonatal infection (sepsis), brain injury (neonatal purulent meningitis or intracranial hemorrhage), maternal history of disease (gestational hypertension, gestational diabetes) and maternal BMI. Parents of toddlers were asked to fill out questionnaires when enrolled to provide sociodemographic profile including ethnicity, siblings, education background and economic income. Information regarding feeding pattern (pure breast milk feeding duration during first 6 months, time of weaning) and nutrition protocol (the usage of donor breast milk, fortification or probiotics) after discharge was obtained via oral answer by telephone at 6 months.

### 2.3. Language and Cognitive Developmental Tests

Language and cognitive developmental outcome of preterm toddlers included was assessed by Bayley Scales of Infant and Toddler Development III (BSIDIII) [9] at 12 ± 2 and 18 ± 2 months CA, respectively. Developmental functioning of cognitive, language and motor skill were determined by item administration under supervision of a trained researcher blinded to the study design. BSIDIII offered standard scores according to different age groups (mean 100 ± 15). Primary outcomes were composite scores of these three domains.

### 2.4. Fecal Sample Collection

Parents of both term and preterm enrolled toddlers were asked to collect fresh stool from diapers using a collecting tube prefilled with 2 mL DNA stabilizer solution (Bio-generating Biotechnology Corp, Changzhou, China) at 12 months, 16 months and 20 months after birth. With full stirring, degradation of DNA can be avoided and the sample collected can be transferred back to the central laboratory at room temperature. Fecal samples were frozen and stored at −80 °C for the DNA extraction procedure.

### 2.5. 16S rRNA Gene Amplicon Sequencing

Fecal DNA extraction was carried out using a TGuide S96 Magnetic Universal DNA Kit (TIANGEN, Beijing, China) following the manufacturer’s instruction book. We chose the V3 and V4 hypervariable region of 16S rRNA gene to complete amplification using 338F (5′-ACTCCTACGGGAGGCAGCA-3′) as a forward primer and 806R (5′-GGACTACHVGGGTWTCTAAT-3v) as a reverse primer. The products were purified, quantified and homogenized to obtain a sequencing library. Qualified libraries were sequenced on an Illumina Novaseq 6000 (San Diego, CA, USA).

### 2.6. Statistical Analyses

For language and cognitive development outcomes, toddlers enrolled were divided into two groups according to information collected: (1) exclusively breast milk fed for a minimum of 3 months during the first 6 months after birth; (2) exclusively formula fed or received a mixture of breast milk and formula for a minimum of 3 months during first 6 months after birth. Neonatal and parental characteristics data were described by mean ± standard deviation (m ± SD), median (interquartile range) or proportion. Independent-sample *t* test or Mann–Whitney *U* test was carried out to compare cognitive, language and motor index scores between toddlers from the breast milk feeding and mixed or formula feeding groups on the basis of distribution. Multiple linear regression was performed to define confounding factors including sex, gestational age, birth weight, neonatal purulent meningitis, intracranial hemorrhage, sepsis, parental educational and income level. All statistical analyses were conducted using SPSS software v21.0 for Windows and *p* < 0.05 was considered statistically significant.

For 16S rRNA gene amplicon sequencing data, raw reads were firstly filtered using Trimmomatic v0.33, then the primer sequences were identified and removed by cutadapt 1.9.1 to generate high-quality reads. Based on overlapping sequences, high-quality reads were assembled by FLASH v1.2.7, then clean reads were generated. Denoising was processed by dada2 [10] in QIIME2 2020.6 [11] to remove chimeric sequences and generate amplicon sequence variants (ASVs). Alpha diversity indices were calculated by the Vegan package in R software and compared using the Student’s *t* test. Principal coordinate analysis (PCoA) based on binary Jaccard distances and Bray–Curtis distances was plotted to describe beta diversity, which was calculated using the Vegan package. Taxonomic annotation of feature sequences was processed by means of Bayesian classifier using SILVA as reference database. QIIME was applied to obtain abundance of each species in samples and distribution histograms at each taxonomic level were generated using R package. The LDA effect size (LefSe) was applied to estimate the effect size of each differentially abundant feature.

We divided assessed preterm toddlers into two groups according to language or cognitive composite scores: good (composite score ≥ 100), below average (composite score < 100). The sample size calculation was based on our preliminary data. The minimum sample size calculated was 46 preterm toddlers (two-sample *t* test, 80% of power in hypothesis test, 0.05 of type 1 error, two-tailed test) to define a difference of 5 points in cognitive, language or motor composite score of BSIDIII (100 versus 95, SD 6) at 18 ± 2 months CA.

## 3. Results

### 3.1. Study Population

In total, 55 of 95 preterm toddlers and 22 term toddlers were enrolled. Neonatal clinical characteristics are summarized in Table 1. For neurodevelopment outcome, 44 preterm toddlers (80%) attended follow-up at 12 ± 2 months CA (11 non-available: 7 unable to contact, 2 outside Beijing, 1 unwilling to come, 1 unable to complete test) and 48 toddlers (87.3%) attended follow-up at 18 ± 2 months CA (7 non-available: 3 outside Beijing, 4 unwilling to come) as shown in Figure 1. Neonatal clinical and sociodemographic characteristics are summarized in Table 2. Participants of the 12 months CA and 18 months CA follow-up study from whom BSIDIII scores were available were comparable on baseline characteristics. For fecal samples, a total of 159 samples from 79 toddlers enrolled (33 samples from preterm toddlers and 18 samples from term toddlers at 12 months after birth, 34 samples from preterm toddlers and 21 samples from term toddlers at 16 months after birth, 34 samples from preterm toddlers and 19 samples from term toddlers at 20 months after birth) were obtained as shown in Figure 2A.

### 3.2. Feeding Group

Sixteen of 44 preterm toddlers (36.36%) followed at 12 months CA and 23 of 48 preterm toddlers (47.9%) followed at 18 months CA received breast milk after discharged. Three of 44 preterm toddlers (6.81%) followed at 12 months CA and 3 of 48 toddlers (6.25%) followed at 18 months CA received formula. Toddlers in the formula or mixed feeding group were fed preterm formula for catch-up growth and then changed to term formula. None of the preterm toddlers enrolled received donor milk. Four of 44 toddlers (9.09%) followed at 12 months CA (1 in breast milk feeding group, 3 in mixed or formula feeding group) and 7 of 48 toddlers (14.58%) followed at 18 months CA (3 in breast milk feeding group, 4 in mixed or formula feeding group) underwent fortification. The basal characteristics reported in Table 1 were compared in preterm toddlers of different feeding groups both at 12 and 18 months CA. No significant difference was detected between the groups (See Table 1 and Table 2).

### 3.3. Neurodevelopment Outcome

As shown in Table 3, at 12 months CA, preterm toddlers in the breast milk feeding group had significantly higher language composite scores than those in the mixed or formula feeding group (86 (79,97) vs. 77 (71.75,79), *p* = 0.008). Though the cognitive composite scores in the breast milk feeding group was better than the mixed or formula feeding group, the difference was not statistically significant (110 (96.25,110) vs. 102.5 (90,110), *p* = 0.507). No significant difference in motor composite score between the two groups was detected (88 (85,91) vs. 88 (85,91), *p* = 0.672).

At 18 months CA, language (106.05 ± 14.68 vs. 90.58 ± 12.25, *p* = 0.000) and cognitive composite scores (107.17 ± 10.85 vs. 99.00 ± 9.24, *p* = 0.007) remained higher in the breast milk feeding group and the differences were significant. There was no significant difference in motor composite score between the feeding groups either (102.04 ± 6.22 vs. 98.00 ± 8.22, *p* = 0.062), as shown in Table 4.

Multiple linear regression was used to assess the association of 12-month CA and 18-month CA BSIDIII scores with the feeding group when gestational age, birth weight, neonatal purulent meningitis, intracranial hemorrhage, sepsis, antibiotics use duration, siblings, maternal BMI, maternal education level and family income were controlled for the analysis. Table 5 reports that the feeding group was the only factor associated with higher language (B = 10.121, 95%CI: 2.54, 17.70, *p* = 0.011) and cognitive composite score (B = 12.46, 95%CI: 3.18, 21.73, *p* = 0.010) at 18 months and language composite score at 12 months CA (B = 8.41, 95%CI: 2.11, 14.71, *p* = 0.010).

### 3.4. Gut Microbiota Diversity and Composition Varied by Sampling Time and Neurodevelopment Outcome

ACE and Shannon indices were used to describe the alpha diversity of gut microbiota. As shown in Figure 2B, at 12 months, the ACE index of the breast milk feeding group and term group was significantly lower than that in the mixed or formula feeding group (*p* = 0.014, *p* = 0.0049, respectively). The Shannon index of the breast milk feeding group and term group was lower than that in the mixed or formula feeding group but the difference was not significant. At 16 months, both the ACE index and Shannon index of term group were significantly lower than that in the mixed or formula feeding group (*p* = 0.0071, *p* = 0.0011, respectively). At 20 months, there were no significant difference in ACE index and Shannon index among three groups. For language score (See Figure 2C), at 12 months, the ACE index and the Shannon index of the good language composite score group were lower than that in the below average group and the difference in the ACE index was statistically significant (*p* = 0.014). At 16 months and 20 months, there was no significant difference in the ACE index and the Shannon index of the good language composite score group compared with the below average group.

PCoA plots were carried out to explore the beta diversity distances of each group. As shown in Figure 3A, at 12 months, the microbiota of the mixed or formula feeding group tended to cluster in left quadrant but the microbiota of the breastmilk feeding group tended to cluster in right quadrant, and a significant difference was found in beta diversity (*R^2^* = 0.196, Adonis *p* = 0.001). At 16 months, the distribution of samples in the breast milk feeding group and term group was more concentrated than that of mixed or formula feeding group and the difference among the three groups was significant (*R^2^* = 0.067, Adonis *p* = 0.020). At 20 months, the samples in the breast milk feeding group and term group scattered around the left diagonal but the samples in the mixed or formula feeding group scattered around the right diagonal, and the difference among the three groups was significant (*R^2^* = 0.063, Adonis *p* = 0.019). The relative abundance of several gut microbiota was analyzed (Figure 3B) and we found *Firmicutes* was the dominate bacteria in three groups, making up 41.29% to 52.48% from 12 months to 20 months. At the genus level, at 12 months, the relative abundance of *Escherichia Shigella* was greatest in the breast milk feeding group and lowest in the mixed or formula feeding group (11.15% vs. 4.55%). At 16 months, the relative abundance of *Veillonella* in the breast milk feeding group (7.67%) and the term group (7.57%) was higher than for the mixed or formula feeding group (4.26%). At 20 months, the relative abundance of *Bacteroides* in the breast milk feeding group (22.83%) and the term group (18.46%) was higher than for the mixed or formula feeding group (10.79%).

As shown in Figure 4A, at 12 months, most of the samples in the good language composite score group distributed in the upper right quadrant and the difference of beta diversity between the good and below average language composite score groups was significant (*R^2^* = 0.249, Adonis *p* = 0.03). At 16 months, the difference of Bray–Curtis distance between samples in the two groups was not significant. At 20 months, the samples in the good language composite score group located in the upward side of the below average language composite score group, and the difference was significant (*R^2^* = 0.136, Adonis *p* = 0.04). As shown in Figure 4B, at the phylum level, the relative abundance of *Firmicutes* was highest in two groups, making up 42.29% to 52.12% from 12 months to 20 months. At the genus level, at 12 months, the relative abundance of *Bacteroides* and *Escherichia Shigella* was greater in the good language composite score group than the below average group (12.65% vs. 6.31%, 12.82% vs. 5.94%, respectively) and the relative abundance of *Bifidobacterium* was lower in the good language composite score group than the below average group (14.11% vs. 32.60%). At 16 months, the relative abundance of *Veillonella* in the good language composite score group was higher than the below average group (8.07% vs. 5.22%). At 20 months, the relative abundance of *Bacteroides* in the good language composite score group was higher than the below average group (23.14% vs. 14.00%).

### 3.5. Linear Discriminant Analysis

Linear discriminant analysis was performed to demonstrate the difference in relative abundance between samples from each group. As shown in Figure 5A, at 12 months, the taxa enriched in the breastmilk feeding group was *Esenbergiella* genus, while the taxa enriched in the mixed or formula feeding group were *Bifidobacterium* breve species and unclassified Bacteroides species (LDA all > 3.5, *p* all < 0.05). At 20 months, taxa enriched in the mixed or formula feeding group was the *Enterobacteriaceae* genus, while the taxa enriched in the breastmilk feeding group was the *Parabacteroides* species (LDA all > 4, *p* all < 0.05). For neurodevelopment outcomes, as shown in Figure 5B, at 12 months, the taxa enriched in the below average language composite score group were *Bifidobacterium* genus and *Faecalibacterium* genus while the taxa enriched in the good language composite score group were *GCA_900066575* genus, *CAG_56* genus and *Citrobacter* genus (LDA all > 4, *p* all < 0.05). At 20 months, the taxa enriched in the good language composite score group was *Bacteriodes fragilis* species while the taxa enriched in the below average language composite score group were *Bifidobacterium* genus and *Collinsella* genus (LDA all > 4, *p* all < 0.05). The taxa enriched in the good cognitive composite score group at 20 months was *Veillonella* genus while the taxa enriched in the below average cognitive composite score group was unclassified *Enterobacteriaceae* genus and *Collinsella* genus (LDA all > 3.5, *p* all < 0.05).

## 4. Discussion

In this cohort study, we checked the neurological development of preterm toddlers at 12 months CA and 18 months CA and we characterized gut microbiota features of preterm toddlers receiving breast milk, mixed or formula feeding and term toddlers of 12 months, 16 months and 20 months after birth. Our data demonstrated that early-stage breast milk feeding practice advantaged both neurodevelopment outcome and gut microbiota profile of preterm toddlers.

Neurodevelopmental outcomes of preterm toddlers have been reported that were negatively affected by poor growth [12]. Formula feeding ensures adequate nutrition intake and mixed feeding mode is common in preterm toddlers because pure breast milk cannot meet the protein and mineral needs of preterm toddlers and providing enough breast milk is also challenging for mothers after premature labor [13]. In previous published findings, the influence of different feeding ways on neurodevelopmental outcomes of preterm toddlers is inconsistent [14,15,16]. Results of our study showed early and exclusive breast milk feeding improved not only language composite scores of preterm toddlers at 12 months CA but cognitive and language composite scores of preterm toddlers at 18 months CA as well.

Results of several previous studies are in accordance with our study reporting associations between an exclusively breast milk-based diet and better neurodevelopment outcomes of preterm toddlers. Patra and colleagues [17] found a dose-dependent relationship between the volume of human milk intake in NICU and improved cognitive outcome at 20 months CA in a very low birth weight (VLBL) infant cohort. Ruys and colleagues [18] reported very preterm infants (gestational age < 32 weeks) fed predominantly (>80%) human milk at term age had better cognitive outcomes at 24 months CA and 8 years. Lapidaire and colleagues demonstrated that no matter what kind of human milk, maternal breast milk or banked donor breast milk, it protects preterm infants against infection/necrotizing enterocolitis (NEC), and the decrease in infection/NEC is associated with better cognitive outcomes at ages 7 and 30 years [19].

Feeding mode in early life is critical to shaping the gut microbiome structure. Our study demonstrated that at 12 months, the ACE index of breast milk feeding group and term group was significantly lower than that in the mixed or formula feeding group. At 16 months, both the ACE index and Shannon index of the term group were significantly lower than those in mixed or formula feeding group. Similar to other studies, formula or mixed feeding is associated with a more diverse fecal microbiota shown by indices of alpha diversity [20]. We also found that breastmilk-fed preterm toddlers had better language and cognitive function than those of mixed or formula-fed preterm toddlers. At 12 months, the ACE index of the good language composite score group was significantly lower than that in the below average language composite group. High alpha diversity was thought to be positive to health. One of the features of adult-like gut microbiota of toddlers is higher alpha diversity and low alpha diversity was reported to be associated with asthma [21] and type 1 diabetes [22]. However, high alpha diversity has also been reported to be found in adult subjects with autism spectrum disorder [23] and major depressive disorder [24]. Similarly, Carlson et al. [25] reported that in healthy term toddlers, higher alpha diversity at 12 months was associated with lower scores on the overall composite scores, visual composite score and expressive language score at 24 months. In their study, the proportion of term toddlers currently receiving breast milk feeding remain highest in the best neurodevelopment outcomes group and the difference was significant (*p* = 0.012). Breast milk feeding duration is one of the major influencing factors of alpha diversity. In our study, most preterm toddlers ceased breast milk feeding after 12 months and for term toddlers, breast milk feeding lasted till 18 months. Compared with the higher alpha diversity group, low alpha diversity may mean that more nutrient resources are taken up by gut microbiota, beneficially impacting language and cognitive development [25] so lower alpha diversity at late infancy is in favor of optimal neurodevelopment outcomes.

We found from 12 to 20 months, that the beta diversity was significantly different among the breast milk feeding group, mixed or formula feeding group and term group. Meanwhile, the difference between beta diversity of the good language composite score group and the below average group was significant at 12 and 20 months. At 12 months, the relative abundance of *Escherichia Shigella* genus was higher in the breast milk feeding group and good language composite group when compared with the mixed or formula feeding group and below average group. Wang et al. reported that compared with formula-fed moderate–late preterm infants, *Veillonella* (18.4%) and *Escherichia Shigella* (15.2%) were the most abundant taxa in mothers’ own breast milk fed infants [26]. In their study, 70% of the preterm infants were delivered by caesarean section and many of them had received a prolonged course of antibiotics. Similarly, Aguilar-Lopez et al. reported that enterotype of preterm infants characterized by a lower diversity and enrichment of *Bacteroides* and *Escherichia Shigella* was related to lower consumption of preterm formula [27]. Furthermore, in their study, *Bacteroides* and *Escherichia Shigella* enriched enterotype was associated with the use of antibiotic and bovine milk fortifiers. In our study, preterm toddlers in the breast milk feeding group also had a relatively longer course of antibiotic usage and this may affect composition of gut microbiota. At 16 months, the relative abundance of *Veillonella* genus in the mixed or formula feeding group was lower than the breast milk feeding group and the term group, in accordance with comparison result of the below average language composite score group with the good group. Likewise, in the CHILD cohort study, Fehr et al. found that *Veillonella* might be one of the species specially provided by breast milk during the period of breastfeeding [28] and Guzzardi et al. reported that abundance of *Veillonella* was associated with better practical reasoning scores at 60 months of age [29]. In our study, the weaning time of preterm infants in the breast milk feeding group was significantly longer than in the mixed feeding group; 8 of 23 preterm toddlers participating in the 18 months CA test in the breast-fed group had not been weaned until 15 months. So, the result of our study supported that *Veillonella* might be one of the species specially related to breast milk feeding. At 20 months, the relative abundance of *Bacteroidetes* genus was higher in the breast milk feeding group than in the mixed or formula feeding group and the same difference was evident in the good language composite group comparing with the below average group. Lefse analysis showed that *Bacteroidetes fragilis* species was enriched in the good language composite group. Jia et al. [30] reported that *Bacteroides* followed the trajectory of a linear increase over time in full-term vaginally delivered infants and another study based on the CHILD cohort reported that healthy term toddlers with *Bacteroidetes*-dominant gut microbiota at 12 months scored higher in the language and cognitive domain of BSIDIII at 24 months [31]. At 20 months, Lefse analysis also showed that *Enterobacteriaceae* genus was enriched in the mixed or formula feeding group and below average language composite score group. The overabundance of *Enterobacteriaceae* has been observed in very low-birth-weight infants [32], indicating the indispensable role of early and exclusive breast milk feeding for healthier gut microbiota composition.

Multiple mechanisms may help to explain the significant improvement of language and cognitive scores of preterm toddlers under the impact of breast milk feeding. Firstly, breast milk contains oligosaccharides. Oligosaccharides interfere with binding of potential microbial pathogens to preterm infants’ intestinal epithelial surfaces thus contributing to decline of infection risk [33]. Meanwhile, oligosaccharides play a vital role in regulation of toddler gut microbiota, serving as a prebiotic that promotes the growth and activity of beneficial gut microorganisms [34], which may influence neurogenesis through the microbiota-gut-brain axis [8]. Secondly, in the breast milk feeding group, the relative abundance of *Bacteroides* increased over time and *Bacteroides fragilis* species could affect the Th1/Th2 balance through capsular polysaccharides and regulate the immune system [35]. Moreover, some *Bacteroides* species contain abundant glycosidase and lyase genes to degrade a variety of complex glycans presented in human breast milk [36]. Thirdly, the precursors of the n-3 and n-6 LC-PUFA present in breast milk promote optimal white matter development and neural growth, so early and exclusive breast milk feeding contributes to the earliest changes in human white matter development [37].

The nutrition protocol has been considered as an opportunity to promote catch-up growth and neurodevelopment among preterm toddlers. On one hand, human milk fortifier is recommended for preterm toddlers fed breast milk and greater weight gain is associated with improved neurodevelopment outcomes [38]. On the other hand, when sufficient breast milk is not available, donor breast milk is a choice [39]. However, only 4 of 44 toddlers (9.09%) followed at 12 months CA (1 in breast milk feeding group, 3 in mixed or formula feeding group) and 7 of 48 toddlers (14.58%) followed at 18 months CA (3 in breast milk feeding group, 4 in mixed or formula feeding group) received fortified breast milk after NICU discharge. None of the preterm toddlers enrolled used donor human milk as supplement. So, the effect of nutrition protocol on long-term neurodevelopment outcomes among preterm toddlers cannot be concluded from our study.

Probiotics was thought to have a chance of preventing neurodevelopmental disorders for neuroactive substances released. *Lactobacillus brevis* and *Bifidobacterium dentium* were reported to increase GABA concentration in vitro in animal studies [40]. Pärtty et al. reported that supplement of *Lactobacillus Rhamnosus GG* (LGG) in infants enrolled can reduce the incidence of autistic spectrum disorder (ASD) and attention deficit and hyperactivity disorder (ADHD) when they grew up [41]. In our study, two preterm toddlers enrolled in the mixed feeding group used probiotics for two months (one during 9–10 months, one during 10–11 months). Six toddlers in the mixed feeding group and two toddlers in the breast milk feeding group were fed with formula supplemented with probiotics. So, in our study, at 12 months, the mixed and formula feeding group had an abundance of *Bifidobacterium breve*. However, at 12 and 20 months, the association between below average language composite scores and abundance of *Bifidobacterium* was found, so this species of *Bifidobacterium* was not effective for better neurodevelopment outcomes. Efforts should be taken to define specific probiotics beneficial for long-term neurodevelopment outcomes and to figure out the mechanisms behind it.

The strength of our study was that this is the first article to profile diet-related gut microbiota and tested diet-related neurodevelopment outcomes in preterm toddlers in the same period and found that early-stage breast milk feeding had beneficial effect in these two aspects through regulation of gut microbiota. There are several limitations of our study. First of all, this is a nonrandomized observational study, thus caution must be taken when generalizing our results. Secondly, only 3 of 55 preterm toddlers enrolled received formula feeding only during the first six months so we merged the formula feeding group and mixed feeding group together to provide more solid evidence. Thirdly, it was not possible to collect stool data in early infancy due to shut down of the hospital clinics for outbreak of COVID-19, and this may bias the results when baseline data were not available. The effect of the feeding method in early life on gut microbiota may have worn out after weaning and many other factors can affect composition of gut microbiota. Although we have controlled relevant clinical and social factors in the multiple linear regression model, our results need to be interpreted with caution. Finally, although we have identified differences in microbiota features such alpha diversity and bacterial taxa, how breast milk feeding may mediate gut microbiota to influence neurodevelopment is unclear. In addition to deepening the insight into toddler gut microbiome to gain better taxonomic resolution, there is a need to conduct longitudinal studies through a more complete period characterizing how breast milk components act synergistically to influence microbial development and neurodevelopment output.

Our study showed that breast milk feeding practice positively affected early life gut microbiome and neurological outcomes in preterm toddlers. Therefore, high-dose breast milk intake should also be prioritized in preterm toddlers with regard to reducing the risks and costs of neurodevelopmental problems in this population.

## 5. Conclusions

In conclusion, we have shown that early and exclusive breast milk feeding of preterm toddlers after NICU hospitalization was beneficial for language development at 12 and 18 months CA and cognitive development at 18 months CA and functioned through regulation of gut microbiota structure for optimal neurodevelopment outcomes.

## Figures and Tables

**Figure 1 microorganisms-11-00866-f001:**
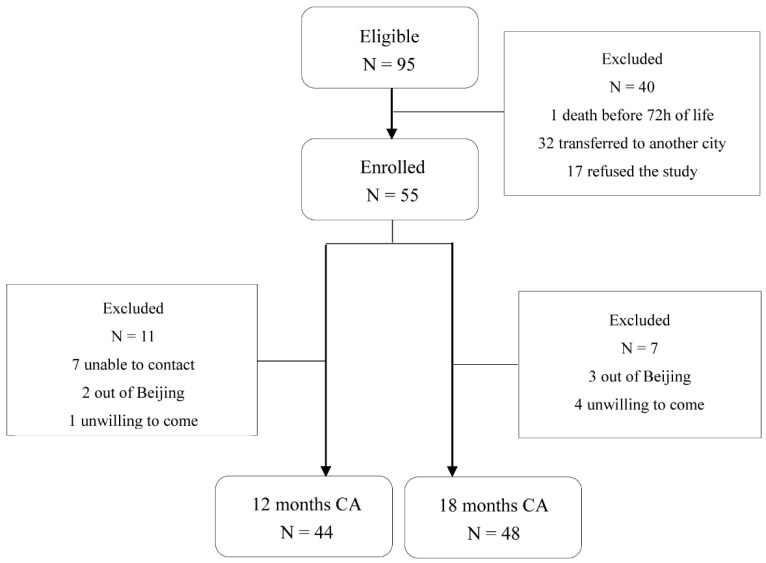
Flowchart of preterm toddlers tested at 12 months and 18 months CA.

**Figure 2 microorganisms-11-00866-f002:**
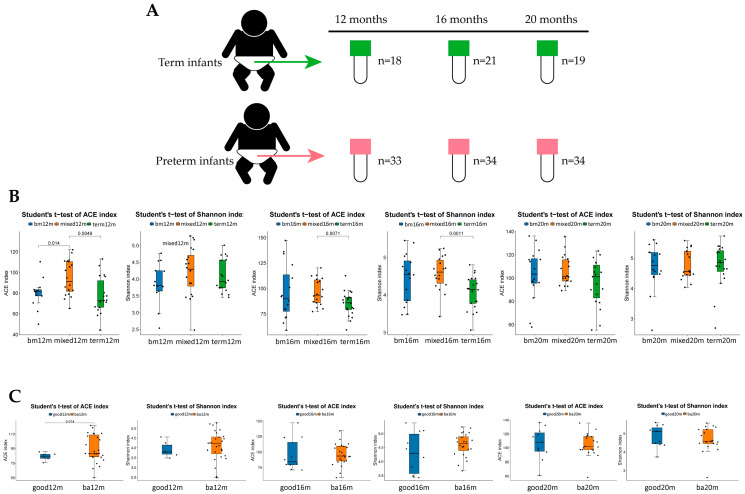
Alpha diversity of each group varied by feeding mode or neurodevelopment outcomes at different sampling times. (**A**) Total of 159 samples collecting at 12 months, 16 months and 20 months; (**B**) ACE index and Shannon index in each group varied by feeding mode at 12 months, 16 months and 20 months; (**C**) ACE index and Shannon index varied by language composite score at 12 months, 16 months and 20 months. Figure legend: bm, breast milk; ba, below average; m, months.

**Figure 3 microorganisms-11-00866-f003:**
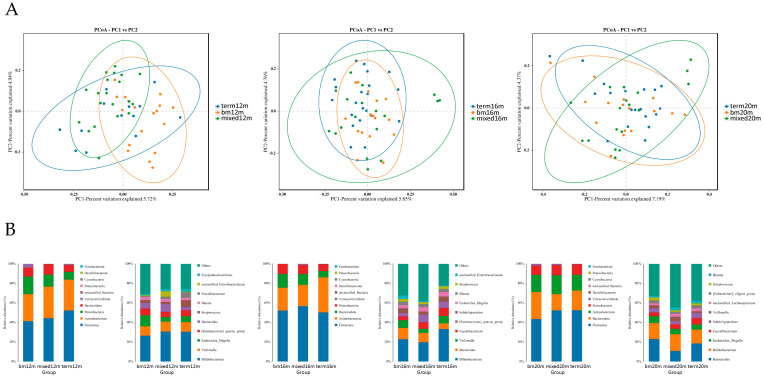
Gut microbiota community structure in each group varied by feeding mode at different sampling times. (**A**) Binary Jaccard distances displayed by PCoA plots at 12 months, 16 months and 20 months; (**B**) relative abundance at phylum level and genus level at 12 months, 16 months and 20 months. Figure legend: bm, breast milk; m, months.

**Figure 4 microorganisms-11-00866-f004:**
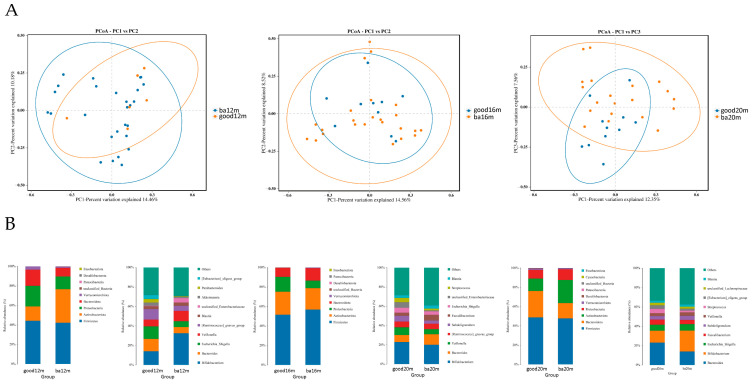
Gut microbiota community structure in each group varied by language composite score at different sampling times. (**A**) Bray–Curtis distances displayed by PCoA plots at 12 months, 16 months and 20 months; (**B**) relative abundance at phylum level and genus level at 12 months, 16 months and 20 months. Figure legend: bm, breast milk; ba, below average; m, months.

**Figure 5 microorganisms-11-00866-f005:**
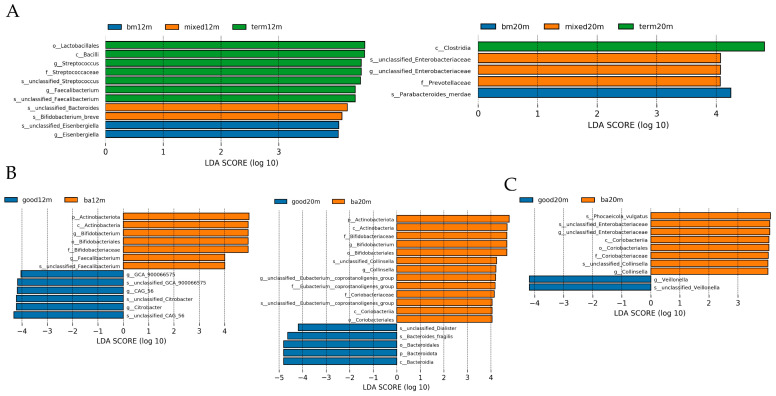
LDA by Lefse illustrating differential bacterial taxa of the gut microbiota community in each group varied by feeding mode or neurodevelopment outcomes at different sampling times. (**A**) differential bacterial taxa varied by feeding mode at 12 and 20 months; (**B**) differential bacterial taxa varied by language composite score at 12 and 20 months; (**C**) differential bacterial taxa varied by cognitive composite score at 12 and 20 months. Figure legend: bm, breast milk; ba, below average; m, months.

**Table 1 microorganisms-11-00866-t001:** Clinical characteristics of preterm toddlers and term toddlers enrolled.

		Preterm Toddlers *n* = 55	Term Toddlers *n* = 22	*p* Value
Male	*n* (%)	27 (49.1)	9 (40.9)	0.516
Gestational age, week	median (IQR)	35 (33–36)	39 (38–40)	0.000
Birth weight, g	mean (SD)	2015.73 (422.78)	3636.95 (500.19)	0.000
Type of feeding				0.000
Breast milk feeding	*n* (%)	25 (45.5)	22 (100)	
Mixed feeding	*n* (%)	27 (49.1)	0 (0)	
Formula feeding	*n* (%)	3 (5.5)	0 (0)	

IQR: inter quartile range; SD: standard deviation.

**Table 2 microorganisms-11-00866-t002:** Clinical and sociodemographic characteristics of study population at 12 and 18 months CA.

		12 Months CA *N* = 44	18 Months CA *N* = 48	*p* Value
Male	*n* (%)	22 (50)	20 (43)	0.423
Gestational age, week	median (IQMR)	35 (34–36)	35 (32–36)	0.213
Birth weight, g	mean (SD)	2117.8 (331.4)	2004.3 (425.6)	0.159
Head circumference, cm	median (IQR)	32 (30, 32.88)	31 (30, 32.5)	0.367
Length, cm	mean (SD)	44.6 (2.9)	43.9 (3.3)	0.264
1 min Apgar score	median (IQR)	10 (9–10)	10 (9–10)	0.153
5 min Apgar score	median (IQR)	10 (10–10)	10 (10–10)	0.107
10 min Apgar score	median (IQR)	10 (10–10)	10 (10–10)	0.107
Hospitalization days	median (IQR)	10 (8–16)	11(8–25.5)	0.200
Caesarean delivery	*n* (%)	35 (79.5)	36 (75)	0.604
Neonatal purulent meningitis	*n* (%)	4 (9.0)	4 (8.3)	0.898
Intracranial hemorrhage	*n* (%)			0.600
Grade 1	*n* (%)	2 (4.5)	3 (6.3)	
Grade 2	*n* (%)	1 (2.3)	2 (4.2)	
Grade 4	*n* (%)	1 (2.3)	1 (2.1)	
Sepsis	*n* (%)	4 (9.0)	11 (22.91)	0.073
Maternal history of disease				
Gestational hypertension	*n* (%)	17 (38.6)	17 (35.4)	0.749
Gestational diabetes	*n* (%)	7 (15.9)	7 (14.6)	0.860
Maternal education				0.871
<12 years	*n* (%)	0	1 (2.1)	
High school	*n* (%)	3 (6.8)	4 (8.3)	
Associate’s degree	*n* (%)	5 (11.4)	4 (8.3)	
Bachelor’s degree	*n* (%)	27 (61.4)	30 (62.5)	
Graduate degree	*n* (%)	9 (20.5)	9 (18.8)	
Average family income, CNY				0.940
CNY 50,000–100,000	*n* (%)	5 (11.4)	7 (14.6)	
CNY 100,000–200,000	*n* (%)	13 (29.5)	15 (31.2)	
CNY 200,000–400,000	*n* (%)	14 (31.8)	12 (25.0)	
CNY 400,000–600,000	*n* (%)	5 (11.4)	7 (14.6)	
CNY >600,000	*n* (%)	7 (15.9)	7 (14.6)	
Feeding mode				0.357
Breast milk feeding	*n* (%)	16 (36.4)	23 (47.9)	
Mixed feeding	*n* (%)	25 (56.8)	22 (45.8)	
Formula feeding	*n* (%)	3 (6.8)	3 (6.25)	
Pure breast milk feeding duration during first 6 months				0.445
Breast milk feeding	median [IQR]	5 (4–6)	5 (4–6)	
Mixed feeding	median [IQR]	0 (0–0)	0 (0–0)	
Time of weaning				
Breast milk feeding	mean (SD)	12 (6)	13 (8)	0.409
Mixed feeding	mean (SD)	10 (6)	8 (6)	0.707

CA: corrected age; IQR: inter quartile range; SD: standard deviation; CNY, Chinese Yuan.

**Table 3 microorganisms-11-00866-t003:** Neurodevelopmental outcomes at 12 months CA according to type of toddler feeding.

12 Months CA	Breast Milk Feeding *n* = 16	Formula or Mixed Feeding *n* = 28	*p* Value
Bayley index score (median (IQR))			
Cognitive	110 (96.25,110)	102.5 (90,110)	0.507
Language	86 (79,97)	77 (71.75,79)	0.008
Motor	88 (85,91)	88 (85,91)	0.672

CA: corrected age; IQR: inter quartile range.

**Table 4 microorganisms-11-00866-t004:** Neurodevelopmental outcomes at 18 months CA according to type of toddler feeding.

18 Months CA	Breast Milk Feeding *n* = 23	Formula or Mixed Feeding *n* = 25	*p* Value
Bayley index score (mean ± SD)			
Cognitive	107.17 ± 10.85	99.00 ± 9.24	0.007
Language	106.05 ± 14.68	90.58 ± 12.25	0.000
Motor	102.04 ± 6.22	98.00 ± 8.22	0.062

CA: corrected age; SD: standard deviation.

**Table 5 microorganisms-11-00866-t005:** Associations of feeding mode, gestational age, birth weight, neonatal purulent meningitis, intracranial hemorrhage, sepsis, family income and maternal education level with language composite score at 12 and 18 months CA and cognitive composite score at 18 months CA.

	Language Composite Score at 12 Months CA		Language Composite Score at 18 Months CA		Cognitive Composite Score at 18 Months CA	
	B (95%CI)	*p*	B (95%CI)	*p*	B (95%CI)	*p*
Feeding mode	10.121 (2.54, 17.70)	0.011	12.46 (3.18, 21.73)	0.010	8.41 (2.11, 14.71)	0.010
Gestational age	−0.94 (−4.84, 2.96)	0.625	−3.40 (−7.39, 5.91)	0.093	0.25 (−2.51, 3.02)	0.854
Birth weight	−0.02 (−0.04, 0.01)	0.150	0.00 (−0.02, 0.03)	0.733	−0.02 (−0.04, 0.00)	0.053
Neonatal purulent meningitis	1.76 (−27.24, 30.76)	0.902	−1.86 (−25.60,21.88)	0.875	10.83 (−5.28, 26.95)	0.181
Intracranial hemorrhage	−4.78 (−11.70,2.16)	0.170	−1.54 (−7.77, 4.70)	0.620	0.60 (−3.73,4.94)	0.780
Sepsis	10.42 (−10.31, 31.15)	0.313	0.97 (−2.19, 4.12)	0.539	0.59 (−1.61,2.79)	0.591
Antibiotics use duration	−0.45 (−1.47, 0.58)	0.378	−0.49 (−1.35, 0.38)	0.263	−0.53 (−1.12,0.07)	0.083
Siblings	−2.08 (−9.80, 5.64)	0.586	2.70 (−6.83, 12.21)	0.570	−1.27 (−7.83,5.30)	0.697
Maternal BMI	−0.87 (−2.06, 0.33)	0.149	−0.66 (2.34, 1.02)	0.432	0.43 (−0.74,1.59)	0.464
Family income	−0.11 (−0.32, 0.10)	0.283	0.04 (−0.23, 0.31)	0.769	0.00 (−0.19, 0.19)	0.996
Maternal education	−1.33 (−6.39, 3.73)	0.595	1.15 (−4.26, 6.56)	0.668	2.49 (−1.23,6.22)	0.183

Results of linear regression analyses, represented as beta with 95% confidence interval (CI). CA: corrected age.

## Data Availability

All data generated or analyzed during this study are available from the corresponding author on reasonable request.

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
