# Peer review of "Regulation of Gut Microbiota through Breast Milk Feeding Benefits Language and Cognitive Development of Preterm Toddlers"

_microorganisms, 2023, doi:10.3390/microorganisms11040866_

Round 1

Reviewer 1 Report

Thank you for the article.

In your aims, you are trying to determine if neurodevelopmental outcomes at 12 and 18 months is associated with mode of feeding during the first 6 months of life and you are corelating the gut microbiome to the mode of feeding and the neurodevelopmental outcomes.

In the first aim, the data is collected retrospectively from medical notes and questionnaire information from the parents. The duration of breast milk given is not stated and the timing of initiation is not given as well. Similarly for the formula fed group. Looking at the data, this is a late preterm group of infants. There is no information as to how these infants were at birth nor of maternal history of disease during pregnancy or mode of delivery, all of which could have an impact on gut microbiota development. There is also no information if the type of formula being given to these infants and whether these were nutritionally fortified/enriched. 

Some of the numbers are confusing as the number of infants who were receiving breast milk increased in the 18 month assessment.

For the second aim of the gut microbiota, the stool samples were only collected at 12, 16 and 20 months which will not reflect the mode of feeding in early infancy but the child's current diet at those ages. The gut microbiota at those ages will be influences also by illnesses, use of antibiotics, etc. It would have been better to collect the stool at early infancy to reflect the impact of the mode of feeding.

The presence of Escherichia shigella is usually pathogenic. The reference is inappropriately interpreted. That paper was looking at very preterm infants and the impact of NICU practices on the stool microbiome, most of which was pathogenic and associated with poor outcomes.

There are several published articles looking at mode of feeding and neurodevelopmental outcomes in the literature.

Author Response

Response to Reviewer 1’s comments:

Question 1: In your aims, you are trying to determine if neurodevelopmental outcomes at 12 and 18 months is associated with mode of feeding during the first 6 months of life and you are corelating the gut microbiome to the mode of feeding and the neurodevelopmental outcomes.

In the first aim, the data is collected retrospectively from medical notes and questionnaire information from the parents. The duration of breast milk given is not stated and the timing of initiation is not given as well. Similarly, for the formula fed group. Looking at the data, this is a late preterm group of infants. There is no information as to how these infants were at birth nor of maternal history of disease during pregnancy or mode of delivery, all of which could have an impact on gut microbiota development. There is also no information if the type of formula being given to these infants and whether these were nutritionally fortified/enriched.

Author Response: We thank the reviewer for this suggestion. We added the data of the duration of breast milk given, 1 min Apgar score, 5 min Apgar score, 10 min Apgar score, mode of delivery, maternal history of disease of infants enrolled to table 2. We added the timing of initiation in line 173 (after discharged). The information of the type of formula being given to these infants was added in line 175-176. The information of infants receiving fortified breast milk was displayed in line 177-180 and discussed in line 411-415.

Question 2: Some of the numbers are confusing as the number of infants who were receiving breast milk increased in the 18 months assessment.

Author Response: We thank the reviewer for the comments. Infants who were receiving breast milk increased in the 18 months assessment because 12 months assessment was closed to New Year holiday and some family went back to hometown so they could not show up in Beijing.

Question 3: For the second aim of the gut microbiota, the stool samples were only collected at 12, 16 and 20 months which will not reflect the mode of feeding in early infancy but the child's current diet at those ages. The gut microbiota at those ages will be influences also by illnesses, use of antibiotics, etc. It would have been better to collect the stool at early infancy to reflect the impact of the mode of feeding.

Author Response: We thank the reviewer for the reminding. However, in early infancy, the outbreak of COVID-19 in China caused a temporary suspension of the research so we failed to collect the stool at early infancy. We have added this part to limitation (line 426 to 428).

Question 4: The presence of Escherichia shigella is usually pathogenic. The reference is inappropriately interpreted. That paper was looking at very preterm infants and the impact of NICU practices on the stool microbiome, most of which was pathogenic and associated with poor outcomes.

There are several published articles looking at mode of feeding and neurodevelopmental outcomes in the literature.

Author Response: We thank the reviewer for this suggestion. We have read published articles looking at mode of feeding of preterm infants and rewritten the discussion part (line 366-373). At 12 months, the relative abundance of Escherichia Shigella genus was higher in the breast milk feeding group and good language composite group when compared with mixed or formula feeding group and below average group. Wang et al. reported that compared with formula-fed moderate–late preterm infants, Veillonella (18.4%) and Escherichia/Shigella (15.2%) were the most abundant taxa in mothers’ own breast milk fed infants. Similarly, Aguilar-Lopez et al. reported that enterotype of preterm infants characterized by a lower diversity and enrichment of Bacteroides and Escherichia-Shigella was related to less consumption of preterm formula.

Reviewer 2 Report

Proper intestinal microbiota immediately after birth should be formed by the bacteria in breast milk and immune factors contained primarily in the colostrum  but also the subsequent phases of lactation. On the microbiota of the fetus and newborn affected by many factors such as maternal and neonatal including mode of delivery. These lymphocytes are activated in the gastrointestinal tract by a number of beneficial bacteria in human milk. Developed by the bacteria normal microbiota can survive for many years, affecting the future-individual health. Extremely important mechanism associated with normal microbiota and the impact on the normal development of CNS activity is the gut-brain axis

Gut -brain axis is the subject of the work presented to me for review. The authors examined the dependence of the language, cognitive developmental and motor  outcome infants on the intestinal microbiota, and thus addressed the issue of gut-brain axis. The introduction describes the basic assumptions and meaning of the gut-brain axis clearly enough. The study is a cohort study regarding  55 preterm newborns< 37 GA and 24 term babies > 37 GA.  Material was divided into two groups: exclusively breast milk fed for a of 3 months during first 6 months after birth; and  exclusively formula fed or a mixture of breast milk and formula for a minimum of 3 months during first 6  months after birth. The flow chard  as well as clinical and sociodemographic characteristics of the material clearly presents the designed studies.Applied methods of both development assessment (III Bayley scale) and genetic identification of bacteria are appropriate and modern. Language and cognitive developmental outcome were assessed at 12 and 18 months of age. Total 159 of stool samples was examined at 12 months, 16 months and 20 months. Gut microbiota community structure in each group varied by feeding mode are beautifully presented graphically. The study study showed breast milk feeding practice positively affected early life gut microbiome and neurological outcomes in preterm toddlers. The research results indicate that in breast milk feeding group  had significantly higher language score than those in mixed or formula feeding1 group .Though the cognitive composite score in breast  milk feeding was better than mixed or formula feeding group, the difference was not statistically significant No significant difference in  motor composite score between the two groups was detected. Extensive and very modern literature. Conclusions short and clearly formulated.  The authors  conclude have  that early and exclusive breast milk feeding of preterm  toddlers after NICU hospitalization was beneficial for language development at 12 and 18  months of CA and cognitive development at 18 months. The work does not have any ethical objections.  All parents of toddlers signed the informed consent form. Discussion was conducted in an interesting way and touches on very important aspects of the changing microbiota in the following months of life. Very rich and current literature. The large number of works come from 2019-2022                  

Author Response

Response to Reviewer 2’s comments:

Proper intestinal microbiota immediately after birth should be formed by the bacteria in breast milk and immune factors contained primarily in the colostrum but also the subsequent phases of lactation. On the microbiota of the fetus and newborn affected by many factors such as maternal and neonatal including mode of delivery. These lymphocytes are activated in the gastrointestinal tract by a number of beneficial bacteria in human milk. Developed by the bacteria normal microbiota can survive for many years, affecting the future-individual health. Extremely important mechanism associated with normal microbiota and the impact on the normal development of CNS activity is the gut-brain axis

Gut -brain axis is the subject of the work presented to me for review. The authors examined the dependence of the language, cognitive developmental and motor outcome infants on the intestinal microbiota, and thus addressed the issue of gut-brain axis. The introduction describes the basic assumptions and meaning of the gut-brain axis clearly enough. The study is a cohort study regarding 55 preterm newborns< 37 GA and 24 term babies > 37 GA.  Material was divided into two groups: exclusively breast milk fed for a of 3 months during first 6 months after birth; and exclusively formula fed or a mixture of breast milk and formula for a minimum of 3 months during first 6 months after birth. The flow chard as well as clinical and sociodemographic characteristics of the material clearly presents the designed studies. Applied methods of both development assessment (III Bayley scale) and genetic identification of bacteria are appropriate and modern. Language and cognitive developmental outcome were assessed at 12 and 18 months of age. Total 159 of stool samples was examined at 12 months, 16 months and 20 months. Gut microbiota community structure in each group varied by feeding mode are beautifully presented graphically. The study study showed breast milk feeding practice positively affected early life gut microbiome and neurological outcomes in preterm toddlers. The research results indicate that in breast milk feeding group had significantly higher language score than those in mixed or formula feeding1 group. Though the cognitive composite score in breast milk feeding was better than mixed or formula feeding group, the difference was not statistically significant No significant difference in motor composite score between the two groups was detected. Extensive and very modern literature. Conclusions short and clearly formulated. The authors conclude have that early and exclusive breast milk feeding of preterm toddlers after NICU hospitalization was beneficial for language development at 12 and 18 months of CA and cognitive development at 18 months. The work does not have any ethical objections. All parents of toddlers signed the informed consent form. Discussion was conducted in an interesting way and touches on very important aspects of the changing microbiota in the following months of life. Very rich and current literature. The large number of works come from 2019-2022

Author response: We thank the reviewer for the comments.

Reviewer 3 Report

Dear authors,

The study is fascinating, but some points need clarification and improvement.
1. I do not understand the division of children concerning the type of feeding method. According to the rule presented, there could be children who were exclusively breastfed for the first three months, and then for the next three months in a mixed way. Which group were they included in?
2 Late preterm infants were included in the study, but even among them, there could be children who were diagnosed with PVL, asphyxia, sepsis, in addition to IVH (which degree?) or meningitis. These are all factors that can adversely affect the development of children. If such complications occurred among the preterm infants studied, they should be included in the confounding factors, certainly, the IVH and meningitis already mentioned in Table 2.
3. Line 171-2: "Three of 44 preterm toddlers (6.81%) followed at 12 months CA and 3 of 48 toddlers (6.25%) followed at 18 months CA received formula."
Line 411: "...only 3 of 55 preterm toddlers enrolled received formula feeding only during the first six months."
Table 2 shows 4 children each were artificially fed.

Author Response

Response to Reviewer 3’s comments:

The study is fascinating, but some points need clarification and improvement.
Question 1: I do not understand the division of children concerning the type of feeding method. According to the rule presented, there could be children who were exclusively breastfed for the first three months, and then for the next three months in a mixed way. Which group were they included in?

Author Response: We thank for the comments. We have added the data of the duration of breast milk given to table 2. The shortest duration of breast milk of the infants in exclusively breastmilk feeding group was 3.5 months and the longest duration of breast milk of the infants in mixed of formula feeding group was 2 months. And we also corrected “at least 3 months” to “over 3 months” (line 22, 29 and 64).

Question 2: Late preterm infants were included in the study, but even among them, there could be children who were diagnosed with PVL, asphyxia, sepsis, in addition to IVH (which degree?) or meningitis. These are all factors that can adversely affect the development of children. If such complications occurred among the preterm infants studied, they should be included in the confounding factors, certainly, the IVH and meningitis already mentioned in Table 2.

Author Response: We thank for this suggestion. Because none of preterm toddlers enrolled were diagnosed with PVL or asphyxia, we have added grade of IVH and neonatal infection rate (sepsis) to Table 2. We have also redone the regression and found feeding mode was still the only influence factor (Table 5).   

Question 3: Line 171-2: "Three of 44 preterm toddlers (6.81%) followed at 12 months CA and 3 of 48 toddlers (6.25%) followed at 18 months CA received formula."
Line 411: "...only 3 of 55 preterm toddlers enrolled received formula feeding only during the first six months."
Table 2 shows 4 children each were artificially fed.

Author Response: We thank for the reminding. The number of toddlers enrolled received formula feeding was three. We have made correction in Table 2.

Reviewer 4 Report

The authors have carried out a good piece of research on the relationship between feeding mode, microbiota and neurodevelopment in preterm infants, for which I congratulate them. However, as the authors themselves indicate, longitudinal studies with a larger population are needed to reach more solid conclusions.

In order to publish the work, the following considerations should be taken into account.

-       If having neonatal purulent meningitis and intracranial haemorrhage can dramatically affect a child's neurological development, why is it not considered grounds for exclusion from the study?

-        Having a high alpha diversity is one of the indicators of a healthy microbiota. In fact, researchers have observed poorly diversified microbiota in individuals with non-communicable diseases and older people. The authors need to explain in more depth, and make a comparative argument with other similar studies, why they consider low alpha diversity to be positive for children's neurodevelopment.

-       It has also been shown that preterm infants have higher nutritional needs than full-term infants, especially for energy and protein. How would these needs be met by exclusive breastfeeding, and is supplementation of breast milk sufficient?

MINOR CORRECTIONS

TABLE 1. The acronyms CA and CNY do not appear in the table, but do appear in the table caption.

Figure 3 does not display correctly

The font size of the figures is too small

I remain at your disposal

Author Response

Response to Reviewer 4

The authors have carried out a good piece of research on the relationship between feeding mode, microbiota and neurodevelopment in preterm infants, for which I congratulate them. However, as the authors themselves indicate, longitudinal studies with a larger population are needed to reach more solid conclusions.

In order to publish the work, the following considerations should be taken into account.

Question 1: If having neonatal purulent meningitis and intracranial haemorrhage can dramatically affect a child's neurological development, why is it not considered grounds for exclusion from the study?

Author Response: We thank for the reminding. We have redone the regression and found feeding mode was still the only influence factor (Table 5). As the sample size of this study is small and the data collection process was really difficult because of repeatedly outbreak of COVID-19 and we found the influence of neonatal purulent meningitis and intracranial haemorrhage on neurodevelopment in our study was acceptable so we didn’t exclude them from our study.

Question 2:  Having a high alpha diversity is one of the indicators of a healthy microbiota. In fact, researchers have observed poorly diversified microbiota in individuals with non-communicable diseases and older people. The authors need to explain in more depth, and make a comparative argument with other similar studies, why they consider low alpha diversity to be positive for children's neurodevelopment.

Author Response: We thank for reviewers’ suggestion. We have read relative articles and made explanation in more depth in discussion part (line 346 to 361). High alpha diversity was thought to be positive to health. However, high alpha diversity has also been reported to be found in adult subjects with autism spectrum disorder and major depressive disorder. Similarly, Carlson et al. reported that in healthy term toddlers, higher alpha diversity at 12 months was associated with lower scores on the overall composite scores, visual composite score and expressive language score at 24 months. In their study, the proportion of term toddlers currently receiving breast milk feeding remain highest in the best neurodevelopment outcomes group and the difference was significant (p=0.012). Breast milk feeding duration is one of the major influence factors of alpha diversity. In our study, most of preterm toddlers ceased brestmilk feeding after 12 months and for term toddlers, brestmilk feeding lasted till 18 months. Compared with higher alpha diversity group, low alpha diversity may mean more nutrients resources taken up by gut microbiota beneficially impacting language and cognitive development so lower alpha diversity at late infancy is in favour of optimal neurodevelopment outcomes.

Question 3: It has also been shown that preterm infants have higher nutritional needs than full-term infants, especially for energy and protein. How would these needs be met by exclusive breastfeeding, and is supplementation of breast milk sufficient?

Author Response: We thank for the question. In our study, the birth weight of 32 preterm infants enrolled were over 2000g and 7 preterm infants enrolled were near 2000g. Breastfeeding initiated after discharge so exclusive breastfeeding could meet the needs of preterm infants enrolled in our study.

Question 4: MINOR CORRECTIONS

TABLE 1. The acronyms CA and CNY do not appear in the table, but do appear in the table caption.

Figure 3 does not display correctly

The font size of the figures is too small

Author Response: We thank for the reviewer’s comments. According to the Reviewer’s comment, i) We deleted CA and CNY in the caption of Table 1. ii) We corrected the result describing part of Figure 3 (line 230 to 232). iii) The font size of the figures has been enlarged.

Round 2

Reviewer 1 Report

Thank you for the revised article. There are still some issues with the paper. The infants studied were divided into breastmilk feeding, mixed feeding and formula feeding based on the duration of the type of feeding in the first 6 months of life. The total duration of the type of feeding is not stated. The timing of weaning also is not stated.

Stool collections were done at 12 months, 18 months and 20 months after birth. By that time, the effect of the initial feeding method will have worn out especially if the breastfeeding has been stopped and weaning has occurred. The microbiota of the infants can be influenced by many factors besides mode of feeding, namely mode of delivery, antibiotics use, maternal weight, etc. probiotic supplementation in early life may also influence the gut microbiota and it is not clear if this information was collected. Also presence of siblings can also affect gut microbiota colonisation. The multiple linear regression used did not correct for some of these relevant factors as well. 

In terms of the microbiota findings, at 12 months old, the breastfed group had an abundance of Esenbergialla sp while the mixed fed and formula group had an abundance of Bifidobacterium breve and Bacteroides species which seems to be contrary to most other findings where Bifidobacterium is probably still the most abundant species especially in the breastfeeding group. How can you explain this? Was it the type of formula used?

Also at 12 and 20 months, you found an association with below average language composite scores with abundance of Bifidobacterium and Faecalibacterium and Bifidobacterium and Collinsella genus. This appears again to be contrary to other studies where presence of Bifidobacterium has been associated with better neurodevelopment. 

In the discussion, some of the conclusions drawn may be misleading. In the Wang paper (line 369), again 70% of the preterms were delivered by caesarean section and many of them had received a prolonged course of antibiotics. They also acknowledge that this can affect the findings on the gut microbiota. In the Aquilqr-Lopez paper(line 371), they descried 2 enterotypes. The enterotype associated with Bacteroides and Escherichia-Shigella was again associated with antibiotic use and the use of bovine milk fortifiers. In Fehr's study (line 377), it was presence of both Bifidobacterium and Veillonella that was associated with better practical reasoning and this was during the period of breastfeeding.

It is known that the gut microbiota changes over time and is affected by multiple factors besides early feeding method. Hence, in analysing the microbiota and trying to corelate it to neurodevelopment, many of these factors need to be taken into consideration.

Author Response

Response to Reviewer 1’s comments:

Question 1: Thank you for the revised article. There are still some issues with the paper. The infants studied were divided into breastmilk feeding, mixed feeding and formula feeding based on the duration of the type of feeding in the first 6 months of life. The total duration of the type of feeding is not stated. The timing of weaning also is not stated.

Author Response: We thank the reviewer for this suggestion. We added the data of the exclusive breast milk feeding duration during first 6 months and the timing of weaning in table 2. We also revised the related part in methods (line 89-90).

Question 2: Stool collections were done at 12 months, 18 months and 20 months after birth. By that time, the effect of the initial feeding method will have worn out especially if the breastfeeding has been stopped and weaning has occurred. The microbiota of the infants can be influenced by many factors besides mode of feeding, namely mode of delivery, antibiotics use, maternal weight, etc. probiotic supplementation in early life may also influence the gut microbiota and it is not clear if this information was collected. Also presence of siblings can also affect gut microbiota colonisation. The multiple linear regression used did not correct for some of these relevant factors as well. 

Author Response: We thank the reviewer for the comments. We controlled antibiotics use, maternal BMI and presence of siblings in the regression model (Table 5) and we also revised related part in methods (line 85-88) and results part (line 202-203). We found feeding group was still the only factor associated with better neurodevelopment outcomes. We have added this part to limitation in discussion. For probiotic use, only two infants in mixed feeding group in our study used probiotics in early life, one infant used probiotics during 9-10 months, the other used probiotics during 10-11 months. We have added this information in discussion (line 432 to 446).

Question 3: In terms of the microbiota findings, at 12 months old, the breastfed group had an abundance of Esenbergialla sp while the mixed fed and formula group had an abundance of Bifidobacterium breve and Bacteroides species which seems to be contrary to most other findings where Bifidobacterium is probably still the most abundant species especially in the breastfeeding group. How can you explain this? Was it the type of formula used?

Author Response: We thank the reviewer for the question. We recollected the brand of formula used among preterm toddlers enrolled and we checked the recipe carefully. We found 6 toddlers in mixed feeding group fed with formula supplemented with probiotics. Two toddlers in mixed feeding group used probiotics during 9-11 months. Three toddlers in breastfed group fed with formula supplemented with probiotics. The type of formula used and probiotic usage in early life in mixed feeding group explained this question. We have added this part to discussion (line 428 to 433).

Question 4: Also at 12 and 20 months, you found an association with below average language composite scores with abundance of Bifidobacterium and Faecalibacterium and Bifidobacterium and Collinsella genus. This appears again to be contrary to other studies where presence of Bifidobacterium has been associated with better neurodevelopment. 

Author Response: We thank the reviewer for the comments. Firstly, Bifidobacterium has been associated with better neurodevelopment attributed to neuroactive substances GABA released. Not every species of Bifidobacterium (as well as other probiotics known) have been proven to release neuroactive substances in vitro. Secondly, not all types of probiotics generate significantly beneficial effects on neurodevelopment outcomes in RCT studies. A meta-analysis of RCT studies showed only supplement of Lactobacillus Rhamnosus GG (LGG) in infants enrolled can prevent ASD and ADHD when they grew up. So the species of bifidobacterium enriched in below average language group was not effective for better neurodevelopment outcomes. We have added this part to discussion (line 423 to 437).

Question 5: In the discussion, some of the conclusions drawn may be misleading. In the Wang paper (line 369), again 70% of the preterms were delivered by cesarean section and many of them had received a prolonged course of antibiotics. They also acknowledge that this can affect the findings on the gut microbiota. In the Aquilqr-Lopez paper (line 371), they described 2 enterotypes. The enterotype associated with Bacteroides and Escherichia-Shigella was again associated with antibiotic use and the use of bovine milk fortifiers. In Fehr's study (line 377), it was presence of both Bifidobacterium and Veillonella that was associated with better practical reasoning and this was during the period of breastfeeding.

Author Response: We thank the reviewer for this suggestion. We have rewritten the discussion part according to reviewer’s comments (line 375-392).

At 12 months, the relative abundance of Escherichia Shigella genus was higher in the breast milk feeding group and good language composite group when compared with mixed or formula feeding group and below average group. Wang et al. reported that compared with formula-fed moderate–late preterm infants, Veillonella (18.4%) and Escherichia/Shigella (15.2%) were the most abundant taxa in mothers’ own breast milk fed infants [26]. In their study, 70% of the preterm infants were delivered by cesarean section and many of them had received a prolonged course of antibiotics. Similarly, Aguilar-Lopez et al. reported that enterotype of preterm infants characterized by a lower diversity and enrichment of Bacteroides and Escherichia-Shigella was related to less consumption of preterm formula [27]. Also in their study, Bacteroides and Escherichia-Shigella enriched enterotype was associated with the use of antibiotics and bovine milk fortifiers. In our study, preterm toddlers in breastmilk feeding group also had a relatively longer course of antibiotic usage and this may affect the composition of gut microbiota. At 16 months, the relative abundance of Veillonella genus in the mixed or formula feeding group was lower than the breast milk feeding group and the term group, in accordance with comparison result of below average language composite score group with good group. Likewise, in CHILD cohort study, Fehr et al. found Veillonella might be one of the species specially provided by breast milk during the period of breastfeeding [28] and Guzzardi et al. reported abundance of Veillonella was associated with better practical reasoning scores at 60 months of age [29]. In our study, the weaning time of preterm infants in breast milk feeding group was significantly longer than in mixed feeding group, 8 of 23 preterm toddlers participating 18 months CA test in breast-fed group haven’t been weaned until 16 months. So the result of our study supported Veillonella might be one of the species specially related to breast milk feeding.

Question 6: It is known that the gut microbiota changes over time and is affected by multiple factors besides early feeding method. Hence, in analyzing the microbiota and trying to correlate it to neurodevelopment, many of these factors need to be taken into consideration.

Author Response: We thank the reviewer for this suggestion. We have tried our best to recollect related information and revised our article carefully based on all suggestions above. We added this part to limitation in discussion (line 456-460).

Reviewer 4 Report

I consider that the manuscript has been significantly improved after  incorporating the reviewers' comments.

However, in my view, there is one issue that remains unclear.

In the paper the authors indicate that the group of preterm infants fed with breast milk, start that type of feeding when they are discharged, but what type of feeding did they receive until discharge? It is unusual for a hospitalized preterm infant to be fed with breast milk, since the nutritional requirements of these infants are very high. During the hospital stay, did they receive formula?
This is very important, as the infant's microbiota is shaped during the first days of life.

The first objective of the work, as copied below, is to find out:

1) whether neurodevelopmental outcomes at 12 and 18 months CA of preterm toddlers receiving exclusive breast milk feeding for over 3 months during first 6 months after birth differ from those who received formula or mixed feeding;

Where it is clearly stated that breastfeeding begins at birth, not at discharge.

If preterm infants have consumed formula milk during hospitalization, the results of the work may change markedly.

I look forward to your clarification

Yours sincerely

Author Response

Response to Reviewer 4

I consider that the manuscript has been significantly improved after incorporating the reviewers' comments.

However, in my view, there is one issue that remains unclear.

Question 1: In the paper the authors indicate that the group of preterm infants fed with breast milk, start that type of feeding when they are discharged, but what type of feeding did they receive until discharge? It is unusual for a hospitalized preterm infant to be fed with breast milk, since the nutritional requirements of these infants are very high. During the hospital stay, did they receive formula?
This is very important, as the infant's microbiota is shaped during the first days of life.

The first objective of the work, as copied below, is to find out:

1) whether neurodevelopmental outcomes at 12 and 18 months CA of preterm toddlers receiving exclusive breast milk feeding for over 3 months during first 6 months after birth differ from those who received formula or mixed feeding;

Where it is clearly stated that breastfeeding begins at birth, not at discharge.

If preterm infants have consumed formula milk during hospitalization, the results of the work may change markedly.

I look forward to your clarification

Yours sincerely

Author Response: We thank for the question. In our study, exlusive breastfeeding initiated after discharge. We added the timepoint to the first objective to avoid confusion: 1) whether neurodevelopmental outcomes at 12 and 18 months CA of preterm toddlers receiving exclusive breast milk feeding initiated after discharge for over 3 months during first 6 months after birth differ from those who received formula or mixed feeding (line 63-65). In methods part and conclusion part we have pointed out exclusive breast milk feeding initiated after discharge (line 91 and 473).